# Seamless Tube-Type Heater with Uniform Thickness and Temperature Distribution Based on Carbon Nanotubes Aligned by Circumferential Shearing

**DOI:** 10.3390/ma12203283

**Published:** 2019-10-09

**Authors:** Yoonchul Sohn, Dongearn Kim, Sung-Hoon Park, Sang-Eui Lee

**Affiliations:** 1Department of Welding & Joining Science Engineering, Chosun University, 309 Pilmun-daero, Dong-gu, Gwangju 61452, Korea; yoonchul.son@chosun.ac.kr; 2Molds & Dies Technology Group, Korea Institute of Industrial Technology, 7-47 Songdo-dong, Yeonsoo-gu, Incheon 21999, Korea; kdu0517@kitech.re.kr; 3Department of Mechanical Engineering, Soongsil University, 369 Sangdo-ro, Donjak-gu, Seoul 06978, Korea; 4Department of Mechanical Engineering, Inha University, Inha-ro 100, Michuhol-gu, Incheon 22212, Korea

**Keywords:** carbon nanotubes, circumferential shearing, alignment, electrical conductivity

## Abstract

The uniform temperature distribution, one of the requirements for long-term durability, is essential for composite heaters. An analytical model for temperature distribution of a tube-type heater was derived, and it revealed that thickness uniformity is one order more important than intrinsic material properties such as density, heat capacity, and electrical conductivity of the heating tube. We introduced a circumferential shearing process to fabricate a flexible, seamless tube-type heating layer of carbon nanotube/silicone rubber composite with outstanding uniform distribution of thickness and temperature, which may be attributed to a shorter characteristic dimension in the circumferential direction than in the axial direction. The temperature uniformity was experimentally verified at various temperatures under heating. The difference in measured thickness and temperature in circumferential direction was within ±1.3~3.0% (for *t_avg_* = 352.7 μm) and ±1.1% (for *T_avg_* = 138.8 °C), respectively, all over the heating tube. Therefore, the circumferential shearing process can be effective for cylindrical heaters, like a heating layer of a laser printer, which fuse toners onto papers during printing.

## 1. Introduction

Carbon nanotube (CNT) composites have been extensively investigated to make use of unique characteristics of CNT such as high electrical and thermal conductivity, robust mechanical property, and fast heating performance. Electrically conductive composites were first demonstrated, taking advantages of the high aspect ratio of the CNTs to form a percolation network [1]. For load-bearing applications, CNT composites with polymers or precursor resins can increase mechanical properties such as stiffness, strength, and toughness [2]. CNT composites are also used to enhance physical properties of carbon fiber composites, which can be primary materials for strong, lightweight wind turbine blades and hulls for maritime security boats [2,3]. Another application is a multifunctional coating material such as multiwalled nanotube (MWNT)-containing paints, reducing biofouling of ship hulls by preventing attachment of algae and barnacles [4]. In lithium ion batteries, small amounts of MWNT powder are blended with active materials and a polymer binder in LiCoO_2_ cathodes and graphite anodes, where CNTs provide both enhanced electrical connectivity and mechanical integrity, leading to an increase in rate capability and life cycles [4,5].

Among the multifunctional applications, CNT composites have also been widely studied in heat-related applications such as patternable micro heaters, temperature sensors, heating glasses for vehicles, thermoelectric devices, water heaters, and flexible de-icing units [6,7,8,9]. By the electric Joule heating (resistive heating) of conducting composites, electrical energy can be converted into heat energy quantitatively. In general, composites that exhibit high conductivity and low heat capacity can be ideal for rapid heating applications. Therefore, CNTs can be attractive filler materials for resistive heating with their high aspect ratios coupled with high electrical and thermal conductivities, as high as 106 S/m and 6600 W/m·K, respectively [10,11,12]. One of the practical applications for CNT resistive heaters is a laser printer fuser [13]. The printer fuser heats up and melts toners electrostatically-attached on a paper so that the toners can be fused on the surface of the paper, while the paper passes between a heating roller and a pressing roller. In the heating roller, a conductive composite as heating element is placed on a polymer support, e.g., polyimide (PI) tube, and a protective and toner-releasing layer such as perfluoroalkoxy alkanes (PFA), as shown in Figure 1a. Coating conductive composites on the PI tube could be formed by extrusion process [13]. However, irregular thickness distribution of the conducting composites through circumferential and axial directions of the roller is often observed, as shown in Figure 1b, which may be attributed to two main causes; high viscosity (or stiffness) of CNT composites, and misalignment between the PI tube axis (or extruded core axis) and the extrusion die axis. 

To obtain uniform thickness in axial and circumferential directions, the alignment between the two axes should be guaranteed. Otherwise, the uneven thickness of the heating element could result in fatal malfunctions by irregular temperature distribution during heating process, as shown in Figure 1c. The problem can be circumvented by using a roll coating process, ensuring uniform thickness of composite materials, leading to enhancement in process repeatability, because the characteristic length of the roll coating process, the diameter of the rollers, is lower than that of the extrusion process, the length of the roller.

In this study, uniform temperature distribution along the circumferential direction, as well as in the axial direction, was achieved by applying circumferential shearing with a two roller system. Viscous pastes were able to be coated on one of the two rollers with a higher rotational speed. The MWNTs were observed to be aligned in the circumferential direction in the circumferential shearing, while the nanofillers are aligned in axial direction in the extrusion process. Fast heating performance was also demonstrated with uniform temperature distribution along both circumferential and axial directions.

## 2. Materials and Methods 

### 2.1. Materials

MWNTs, purchased from Hanhwa Chemicals (Ulsan, South Korea), were 100–200 μm and 10–20 nm in length and diameter. Silicone rubber used as polymer matrix was Sylgard 184, a type of polydimethylsiloxane (PDMS), purchased from Dow Corning (Midland, MI, USA). The combination of the two materials was chosen for flexibility and elastomeric response for efficient toner fusing during printing [13], and also for material stability at least up to 230~250 °C [14,15,16].

### 2.2. Fabrication of MWNT/Silicone Composite and Printer Fuser

For the dispersion of MWNTs into PDMS, a two-step process was applied; a planetary milling as preliminary first dispersion process, and then a three-roll milling (TRM) as primary dispersion process. At first, MWNTs and PDMS were weighed and then manually mixed. The mixing ratio of the elastomer base and the curing agent of PDMS was 10:1 by weight. The mixture was planetary-milled and then three-roll milled. 

Effects of the concentration of MWNTs on electrical conductivities of the MWNT/PDMS composites were evaluated in order to satisfy required axial electrical conductivity of the seamless composite heating tube for a given thickness. In the evaluation, the volume fraction of MWNT and PDMS were calculated by assuming their densities to be 1.85 g/cm^3^ and 1.03 g/cm^3^, respectively. 

A circumferential shearing machine was designed to coat the MWNT/PDMS pastes onto a hollow cylindrical PI tube, as shown in Figure 2 and Appendix A. The diameter and the length of the rolls were 24 mm and 300 mm, respectively. The 50 μm thick polyimide tube was placed onto the 1st roller. The 1st and the 2nd roller angular speeds, *v_1_* and *v_2_*, were set up to 300 and 100 rpm, respectively, based on the evaluation of film thickness uniformity, which is shown in Appendix A. Due to the speed difference of the two rollers, the MWNT pastes fed between the two rollers (Figure 2a) were rolled up onto the 1st roller at the higher rotational speed (Figure 2b), and then a uniform composite film on the 1st roller was obtained by separating the 2nd roller from the 1st one (Figure 2c).

In order to build up a printing fuser, metallic electrodes were deposited on the ends of the heating element to apply voltage, and then the MWNT/PDMS heating tube was covered together with a toner release layer of perfluoroalkoxy (PFA) tube, which was 50 μm in thickness, as shown in Figure 2d. The assembled printing fuser was cured at 100 °C at 30 min. and then 180 °C at 2 h. The exposed region of the electrode had a good contact with terminals for power supply, even during roller rotation. A pressing roller was assembled with the heating roller to supply sufficient both heat and pressure to toners on papers, as shown in Figure 1a.

### 2.3. Characterization

Microstructures of MWNT/PDMS composites on the PI tube were observed by scanning electron microscope (SEM) (model: UHR-SEM S-5500, Hitachi, Tokyo, Japan). Electrical conductivity of MWNT/silicone composites was measured according to IEC Standard 93 method (VDE 0303, Part 30) using two-point source meter. Polarized Raman analysis with an excitation wavelength of 532 nm was made to evaluate degree of alignment the MWNTs in the axial and circumferential direction of heating tubes. For the observation of electric heating characteristics of the seamless heating tube, direct current (DC) was applied to the printer fuser with a power supply (NI PXI-1033, National Instruments, Austin, TX, USA). Temperature distribution over the fuser was recorded and imaged by an infrared (IR) camera (Advanced Thermo, TVS-500, Tokyo, Japan).

## 3. Results and Discussion

### 3.1. MWNT/PDMS Pastes

Carbon nanotubes used in this study are shown in Figure 3a. As-received MWNTs consisted of MWNT bundles. Diameters of the bundles were in scales of several micrometers, while some of them were over 10 μm. In a close observation of MWNT bundles, the MWNTs are entangled with each other from van der Waals interaction. The conductive fillers, MWNTs were dispersed in the elastomer, PDMS, with the two-step dispersing process using a planetary mixer and three roll mill (TRM). 

For the TRM process, the repetition number of the TRM process and the pressure between the rolls were varied to obtain maximum conductivity of the composites by optimizing dispersion of the MWNTs in the silicone rubber [16]. Electrical conductivity of MWNT/silicone composites increased as the number of TRM increased and reached the maximum value after six times of repetition in our previous study [16]. Therefore, the composites which had the same TRM processing times were used for the fabrication of MWNT/PDMS paste. The MWNTs are observed to be in excellent degree of dispersion in the polymer matrix, as presented in Figure 3b. 

### 3.2. Analytic Modelling for Temperature Uniformity of Tube-Type Heater

As shown in Figure 1a, the heating roller consists of a conductive heating element between the PI tube and PFA protective layer. To estimate the effects of thickness and electrical conductivity of the heating element on temperature uniformity under an applied voltage, an analytic model was derived for the heating tube, as presented in Figure 4. Axial and circumferential directions of the tube are denoted as *z* and *θ*, respectively. Thickness and outer radius of the heating element are denoted with *t* and *R_o_*, respectively. 

There are assumptions for this model:Heat loss and heat transfer on tube surface are assumed to be disregarded.MWNTs are evenly dispersed, and the degree of MWNT alignment is uniform in the composite tube, which means that electrical conductivity of the (*i*,*θ*) element, *σ_i,θ-A_* = *σ_A_*, and *σ_i,θ-C_* = *σ_C_*, where *A* and *C* stand for axial and circumferential, respectively.Since the electrodes are located at both ends of the heating tube, the total electrical current (*I*) flows through the axial direction. Based on that, the current (*I_θ_*) of an axial element for the tube length (*L*) and a circumferential length (*R_o_*·Δ*θ*) are assumed to be the same, that is, *I_θ_* = *Σ*_i_(*I_i,θ_*) = *I*/(Δ*θ/2π*), where *I_i,θ_* is the current flowing in the (*i*,*θ*) element in the axial direction. This assumption is reasonable in case the average values of thickness and electrical conductivity of the axial elements are approximately same.The *i*-th section (circumferential element) is electrically connected in series, and the electrical current running through the *i*-th section (*I_i_*) is constant, that is, *I_i_* = *Σ_θ_*
*(I_i,θ_)* = *I*.(*i*,*θ*) elements with a cross sectional area of *t_i,θ_* × *R_o_*Δ*θ* are electrically connected in parallel, and each element experiences the same voltage drop along Δ*z*, the length of the *i*th section, that is, *V_i,θ_* = *V_i_*, where *V_i_* is the voltage drop in the *i*-th section.

Electrical resistance of the (*i,θ*) element is expressed in Equation (1).
(1)$$r_{i,\theta }=\frac{\Delta z}{R_{o}\Delta \theta \cdot t_{i,\theta }}\cdot \frac{1}{\sigma_{i,\theta }}$$

From energy conservation, thermal energy of the heating tube is the same as electrical energy applied. *σ_i,θ_* is electrical conductivity of (*i,θ*) element.
(2)$$m_{i,}c_{i,}\Delta T_{i,\theta }=\frac{{\left(V_{i,\theta }\right)}^{2}}{r_{i,\theta }}t$$

Equation (2) can be rewritten, together with Equation (1).
(3)$$\left(\rho_{i,\theta }\cdot t_{i,\theta }\cdot R_{o}\Delta \theta \cdot \Delta z\right)c_{i,\theta }\Delta T_{i,\theta }=\frac{{\left(I_{i,\theta }\cdot r_{i,\theta }\right)}^{2}}{r_{i,\theta }}t$$
where *m*, *c*, and *ρ* represent mass, heat capacity, and density of the heating tube, and t is the time during which the electric power is supplied. Based on the 2nd assumption, *ρ_i,θ_* = *ρ* and *c_i,θ_* = *c*, respectively. The heating rate or thermal response, Δ*T_i,θ_*/*t*, can be expressed in Equation (4).
(4)$$\frac{\Delta T_{i,\theta }}{t}=\frac{I_{i,\theta }{}^{2}}{\rho \cdot c\cdot {\left(R_{0}\Delta \theta \right)}^{2}}\cdot \frac{\sigma_{i,\theta }}{{\left(t_{i,\theta }\right)}^{2}}$$

Therefore, the temperature at a specific point (*i,θ*) of the heating tube depends not only on intrinsic material properties of MWNT/PDMS composite such as electrical conductivity, density, and heat capacity, but also on extrinsic parameters such as thickness and diameter of the heating tube. The temperature is linearly dependent on the former, while it is dependent on the square of the latter geometric variables, revealing that the latter is more dominant for temperature uniformity and heating rate. A process conducted to circumferential direction is expected to have an advantage on easily controlling the geometrical parameters compared to those done to axial direction like extrusion process, because a characteristic dimension can be short in the circumferentially-controlled process, that is, circumferential length, 2π*R_o_*, is shorter than axial tube length, *L*. Therefore, the circumferential shear rolling process was adopted in this study.

In addition, in terms of material properties for the heating element, low heat capacity and high electrical conductivity are beneficial for fast heating. Therefore, carbon nanotubes can be the best candidate with such low specific heat of 0.75 J/g·K for MWNT [17] and high conductivity of 10^4^~10^5^ S/m [18].

### 3.3. Printer Fuser Fabricated by Circumferential Shearing Process

A PI tube with copper electrodes at the tube edges was inserted on a faster roller before the paste was fed. In this study, the rolling speed ratio of the 1st and 2nd rollers varied from 1.1 to 4.5, and the optimum ratio, 3.0, was set up, on the basis of film thickness uniformity (see Appendix A). When the ratio was too low and high, irregular circumferential patterns with thickness differences on composite surfaces were formed, while there was a uniform surface in the optimized condition, as shown in the supporting information. After the coating process of MWNT/PDMS composite was completed, a PFA protective film was prepared on the heating tube. The final structure of the printer fuser consists of PI/heating layer/PFA with the copper electrodes on both ends of the tube, as shown in Figure 2d. For the secure contact of the electrodes and the MWNT/SR heating layer, a silver paste was applied at the edge of the composite.

The MWNTs are aligned in the circumferential direction during the shearing process, because the MWNT paste was sheared to the circumferential flow direction. On the other hand, the nanotubes are preferentially aligned in the axial direction in the extrusion process for the same reason. The filler alignment during the coating process is of great importance, since it strongly affects the electrical conductivity of the composite between electrodes, determining the total resistance of composite heaters. Equation (4) reveals that the resistance of the heating tube depends more on thickness difference than on the electrical conductivity, which is a main reason why the circumferential shearing process was introduced in this study. 

Figure 5 shows the thickness variation of the tube-type heating tube in both the axial and circumferential directions, along which 11 and 6 samples were prepared, respectively, as shown in Figure 5a. 

For a given axial position, the maximum thickness differences (|*t_max_* − *t_min_*|*_max_*) are within ±1.3~3.0% for the average temperature in circumferential direction. For the overall average thickness of 357.9 μm (*t_avg_all_*) over the whole area of the heating tube, all the measured thicknesses lay within 339~374 μm (±5% of the average), as shown in Figure 5b,c. In the axial direction of the tube, the thicknesses around the center of the tube were higher than those near the edges, as shown in Figure 5b. This may be attributed to the feeding position of the MWNT paste, as it was fed at the center of the PI tube. 

In the circumferential direction, the average thicknesses in the axial direction for a given circumferential position range between 353.4 µm (*C*#01) to 361.9 µm (*C*#04), close to *t_avg_all_*, as shown in Figure 5c,d. Based on the analytical model for temperature uniformity in Equation (3), the outstanding uniform thickness in the axial and circumferential directions can guarantee temperature uniformity on the seamless heating tube in both directions, together with the uniform dispersion of MWNTs and the even shear rate along axial direction.

In addition, the thickness discrepancy between the center and the edges of the heating tube can be expected to be further minimized by adjusting the number of feeding positions and the amount of the MWNT paste.

### 3.4. Anisotropic Characteristics of Tube-Type Heating Element

Electrical conductivity of the two-roller processed MWNT/PDMS printer fuser was measured as a function of MWNT concentration and alignment direction, as presented in Figure 6a. During the two-roller coating process, MWNTs are aligned in the circumferential direction of the heating tube. Therefore, in all the specimens with varying MWNT contents, electrical conductivities in the circumferential direction are always higher than that in the axial direction. 

Ratios of electrical conductivities in the circumferential to axial directions are presented in Figure 6b. With a very small amount of MWNTs, 0.1 wt%, the anisotropy of electrical conductivity was as high as ~57. It may be attributed to the fact that the MWNT content was so low that they could not be connect with each other along the axial direction, while connected along preferential alignment (circumferential) direction [18]. With more than 1 wt% of MWNTs, the anisotropy varied between 2.7 and 1.9. The ratio gradually decreased with increasing MWNT contents, as shown in Figure 6b.

The anisotropy of electrical conductivity resulted from preferential alignment of MWNTs in the circumferential direction [18]. For quantitative analysis of MWNT alignment after the two-roll coating process, polarized Raman spectroscopy was measured and the results are presented in Figure 7. Intensities of polarized Raman spectra are summarized in Table 1.

The two main typical graphite bands are present in the Raman spectrum of MWNT bundles: The band around 1580 cm^−1^ (G band) assigned to the in-plane vibration of the C–C bond, i.e., sp^2^ configuration, and typical of defective graphite-like materials, and the band around 1350 cm^−1^ (D band) activated by the presence of disorder in carbon systems. For the D and G bands, *I_D_*/*I_G_* ratios were in the range of 1.02 to 1.10, regardless of direction and MWNT contents. In this study, little change in the *I_D_*/*I_G_* ratio may be attributed to use of the same quality of CNTs in all the specimens after the dispersion process.

Change in the peak intensity with varying polarization direction was clearly identified in Figure 7 and Table 1. For the composite with MWNT 1 wt% in Figure 7a, the Raman peak intensity from circumferential direction was much larger than that from axial direction. Both I_D_ and I_G_ increased up to three times, when polarization direction changed from the axial direction to the circumferential direction, that is, *I_G,circum_/I_G,axial_* and *I_D,circum_/I_D,axial_* are ~3, as shown in Figure 7a. In the same way, intensity ratios for the two directions increased up to two times for the composite with MWNT 7.3 wt%. For the composite in Figure 7b, it is obviously shown that Raman peak intensities are increasing, and both *I_D_* and *I_G_* increased as angles increase from axial (0°) to circumferential (90°) direction, and *I_G,circum_/I_G,axial_* and *I_D,circum_/I_D,axial_* are ~2, which is consistent with the preferential alignment of MWNTs in the circumferential direction during the two-roll coating process, as indicated by the anisotropic electrical conductivity in Figure 6b. The anisotropy of conductivity (*σ_circum_/σ_axial_*) were 2.7 (at 1 wt%) to 2.2 (at 7.3 wt%), which were comparable to *I_G,circum_/I_G,axial_* or *I_D,circum_/I_D,axial_*, 3 and 2, respectively, for the corresponding MWNT contents. 

Anisotropic physical properties of MWNT composites have been investigated. Ra, et al. [19] fabricated MWNTs-embedded polyacrylonitrile (PAN) nanofiber paper by electrospinning process. Electrical conductivity of the carbonized MWNT/PAN nanofiber papers was highly anisotropic, i.e., the conductivity parallel to the winding direction is about three times higher than that perpendicular to the winding direction. Inoue, et al. [20] fabricated MWNT papers from MWNT webs, and reported high anisotropy ratios of 7.3 in resistivity and of 8.1 in thermal conductivity due to the high alignment of the ultra-long MWNTs having lengths of millimeters. In the polarized Raman spectra, for the G band, the parallel polarization intensity was 4.4 times higher than the perpendicular polarization intensity.

In Figure 7b, peak positions of D and G band maximums were red-shifted when polarization direction moved from axial direction to circumferential direction: 1352.9 to 1348.7 for the D band maximum, and 1586.3 to 1581.2 for the G band maximum. L. Bokobza and J. Zhang [21] reported that weak interaction between MWNTs dispersed in the composite resulted in blue-shift of Raman spectra, compared with the composite with bundled MWNTs having stronger interaction between the conductive nanoparticles. In this study, stronger interaction between MWNTs in the circumferential direction is expected due to the preferential alignment of MWNTs, which may contribute to the redshift of Raman spectra.

### 3.5. Heating Performance of Printer Fuser

Heating performance of the printer fuser fabricated was demonstrated as presented in Figure 8a. Uniform temperature distribution can be achieved by the optimal dispersion and even alignment of MWNTs, and the uniform thickness of the heating tube. During the heating process, infrared camera images were taken to identify temperature distribution of the printer fuser. The images of temperature distribution, taken at 1.9, 3.8, and 5.2 s, are presented in Figure 8b–d, respectively. The electrical heating was ceased at controller setting temperature of 200 °C, which resulted in maximum tube temperature of 210 °C at 5.8 s, followed by temperature drop with air cooling. Figure 8e is the temperature distribution along the circumferential direction at the positions #1 to #3 in the Figure 8c. For each position, average temperature values with standard deviation in the circumferential direction were 135.7 (±0.6), 142.2 (±0.7), and 138.8 (±0.8) °C, while differences between the maximum and minimum temperatures, (*T_ma_*_x_-*T_min_*), in circumferential direction were 1.7, 2.3, and 3.0 °C, respectively. 

Figure 8f is the temperature distribution in axial direction at the three moments in the Figure 8b–d. The images clearly show that the MWNT/PDMS heating tube possesses the uniform temperature in circumferential and axial directions. Temperature decreases from center to edges in the axial direction.

As the heat transfer was under transient state, generated heats were still conducted to the other parts of the heating and pressing rollers in Figure 8. Therefore, the temperature distribution is obviously enhanced in steady states at a specific operating temperature. The uniform temperature distribution is attributed to uniformity in thickness and electrical conductivity, as analyzed by Equation (3).

Note that the fusing system having MWNT/PDMS composite as heating element reached 180 °C (melting temperature of toner) in 5.0 s and 200 °C in 5.5 s. With this fast heating performance, it can be realized to decrease the first print-out time, which is meaningful for on-demand printing that papers are printed out as soon as customers request printing.

## 4. Conclusions

A flexible, seamless tube-type heater having uniform temperature distribution and fast thermal response was developed by using a newly-proposed circumferential shearing. Important variables for heating performance, temperature uniformity, and heating speed, were analytically formulated for the composite heating tube. From the theoretical analysis of temperature distribution on the cylindrical composite tube, as shown in Equation (4), the morphological configuration of the heating tube, such as composite thickness and tube diameter, turned out to be one-order more essential for the heating performance than intrinsic material properties such as density, heat capacity, and electrical conductivity. Based on the analytical model, we introduced the circumferential shearing process to fabricate a printer fuser with uniform temperature distribution in the circumferential direction, as the process can provide uniform thickness and shear rate. The circumferential shearing can be a good selection for seamless composite tube structures with outstanding thickness uniformity, especially for the composite paste with high viscosity, which makes difficulty in the fabrication process, such as extrusion.

## Figures and Tables

**Figure 1 materials-12-03283-f001:**
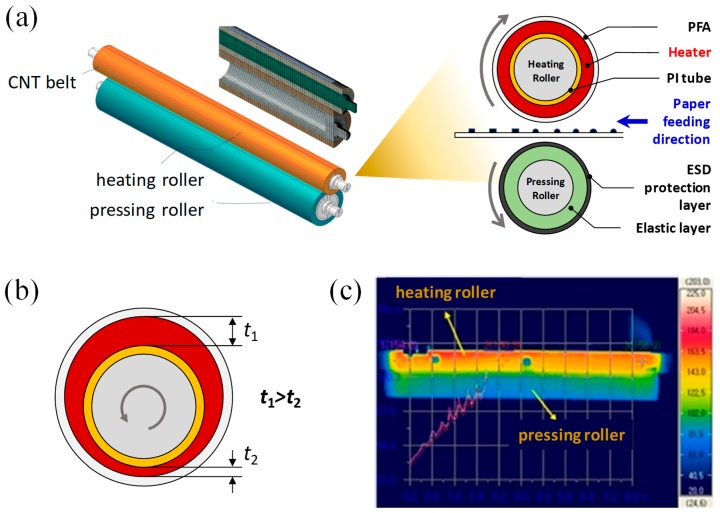
(**a**) Schematic of laser printer fusing element, (**b**) thickness discrepancy of multiwalled nanotube (MWNT)/rubber heater layer in circumferential direction caused by misalignment of the roller axis to extrusion axis, and (**c**) extruded heating roller showing temperature irregularity in the circumferential direction.

**Figure 2 materials-12-03283-f002:**
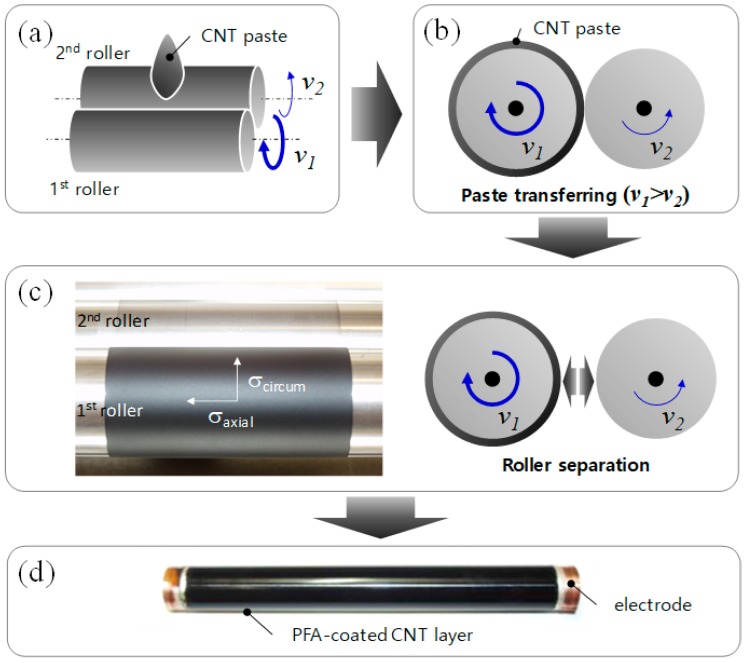
Circumferential shearing process for uniform coating on a cylindrical polyimide (PI) tube. (**a**) MWNT paste feeding, (**b**) paste transferring on a roller with a higher speed, (**c**) roller separation, and (**d**) MWNT heating element with a perfluoroalkoxy alkanes (PFA) layer and electrodes.

**Figure 3 materials-12-03283-f003:**
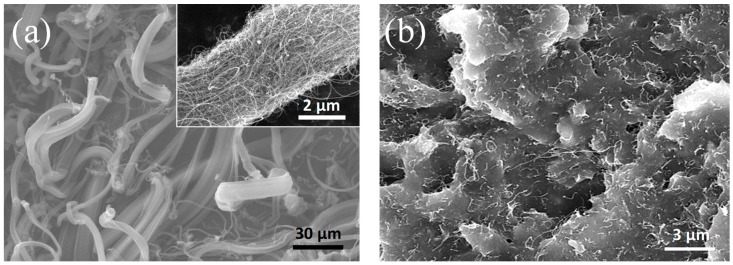
SEM micrographs of (**a**) as-received MWNTs, and (**b**) MWNT/PDMS composite.

**Figure 4 materials-12-03283-f004:**
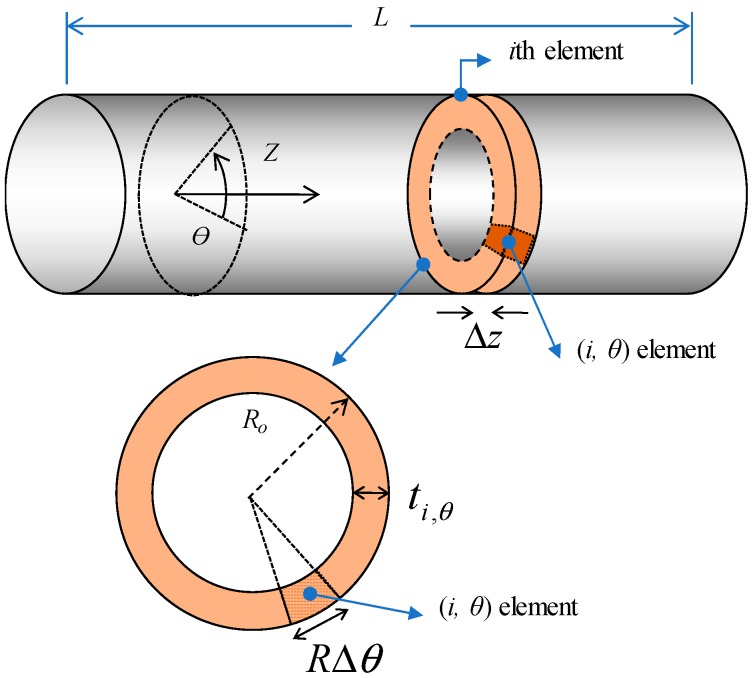
A schematic of tube-type heater for analytical model of temperature distribution.

**Figure 5 materials-12-03283-f005:**
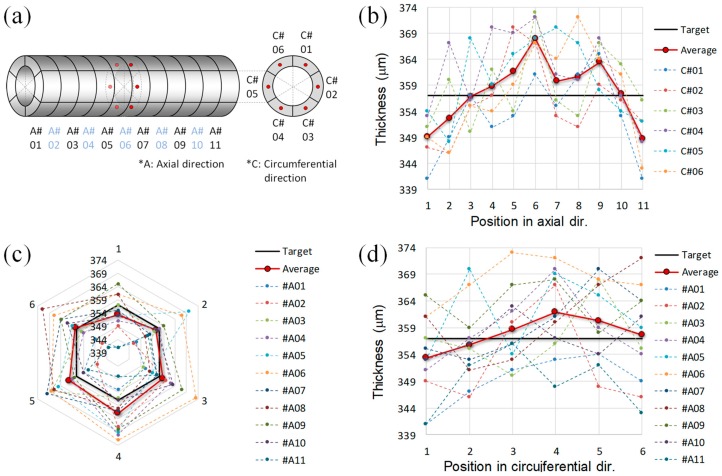
Analysis of thickness distribution. (**a**) denotation of subsection of composite tube, and measured thicknesses in the axial direction (**b**), and in the circumferential direction (**c**,**d**).

**Figure 6 materials-12-03283-f006:**
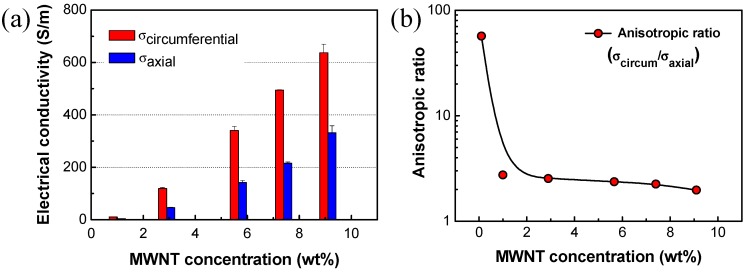
(**a**) Electrical conductivity of MWNT/PDMS composite. (**b**) Anisotropic ratio of electrical conductivity as a function of MWNT contents (axial: axial direction, circum: circumferential direction).

**Figure 7 materials-12-03283-f007:**
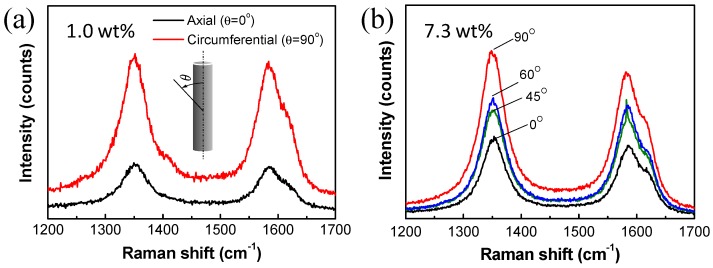
Polarized Raman spectra of MWNT/PDMS composites with MWNT content of (**a**) 1.0 wt%, and (**b**) 7.3 wt%.

**Figure 8 materials-12-03283-f008:**
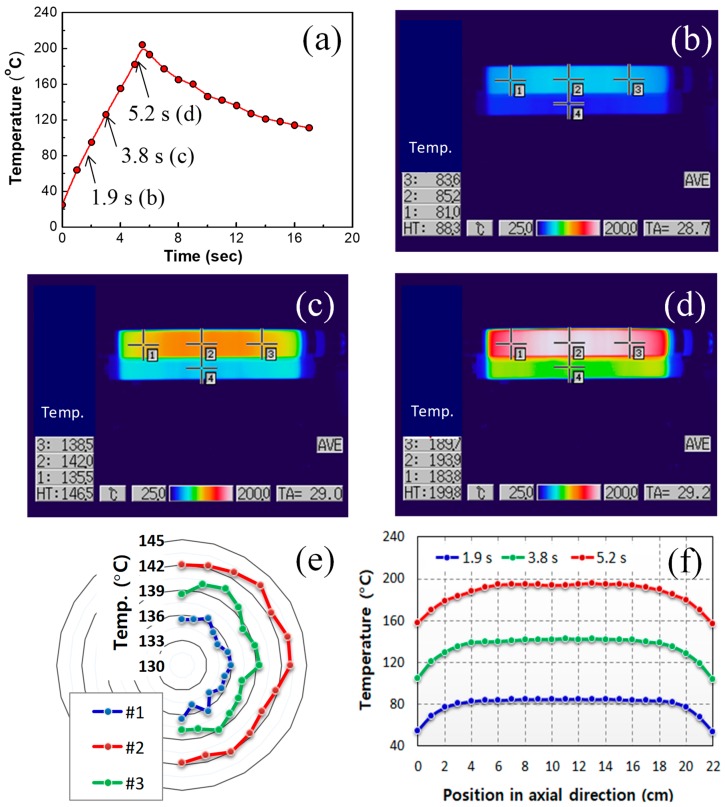
(**a**) Temperature change of MWNT/PDMS fuser with increasing time. Infrared camera images at heating time of (**b**) 1.9 s, (**c**) 3.8 s, and (**d**) 5.2 s. Temperature distribution (**e**) in the circumferential direction at 3.8 s for the three points in Figure 8c, and (**f**) in the axial direction at each moment (1.9 s, 3.8 s, and 5.2 s).

**Table 1 materials-12-03283-t001:** Polarized Raman intensities of MWNT/PDMS composites by circumferential shearing process.

MWNT Content	Polarization Angle	*I_D_*/*I_G_*	*I_D_*/*I_D0_* ^1^	*I_G_*/*I_G0_* ^1^
1.0 wt%	90°	1.05	2.95	3.02
	0°	1.07	-	-
7.3 wt%	90°	1.14	2.04	2.01
	60°	1.07	1.47	1.55
	45°	0.92	1.32	1.63
	0°	1.12	-	-

^1^
*I*_D0_ and *I*_G0_ represent peak intensities extracted from axial direction (*θ* = 0°).

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
