# Peer review of "Seamless Tube-Type Heater with Uniform Thickness and Temperature Distribution Based on Carbon Nanotubes Aligned by Circumferential Shearing"

_materials, 2019, doi:10.3390/ma12203283_

Round 1

Reviewer 1 Report

Some (10) corrections.

1. Line024: A mistyping (letters) error; modify 'din ensity' to be:
'than intrinsic material properties such as density, heat capacity, and electrical conductivity of the'

2. Line108: Add the word "angular" in 'speeds' to be "angular speeds", to be:
"roller. The 1st and the 2nd roller angular speeds, v1 and v2, were set up to 300 and 100 rpm, respectively, based"

3. Line131: Add the frequency (50 ? or 60 Hz) for the (experimental) electrical heating, in:
"heating characteristics of the seamless heating tube, alternating current (?) was applied to the printer"

4. Line217: a) If you used Ni electrodes then modify the incompatible (Cu like) colour of the two-electrodes in Figure 2d.
Line218: b) Correct (4->2) numbering reference: 'Figure 4d' to be "Figure 2d".

5. Line250: Clarify, how did you measure the electrical conductivity, i.e. say, clearly, if it is a two-electrodes' configuration, DC or AC-conductivity (frequency= ? Hz):

6. Line272: Add the Laser's frequency for the Raman spectroscopy, in:
coating process, polarized Raman spectroscopy was measured and the results are presented in Figure 7

7. Line320: Note/add the (temperature) error values in this temperature (180oC) range. Note errors of your temperature meter(s) taken from its specifications, in:
"circumferential direction were 1.7, 2.3, and 3.0 oC, respectively. These correspond to +-0.65, +-0.8, and"

8. Line326: If electrical heating is, actually, interrupted at time about 5.8s, then make it more clear, either adding a note in the text or/and the graph of Figure 8a.

9. Line343: Modify 'structural configuration' to be "morphological configuration"; so, it might be:
'for the composite heating tube. From the analysis, the "morphological configuration" of the heating tube'

10. Line344: Clarify more, how the conclusion wording 'one-order' comes from (which?) experiments, in:
'such as composite thickness and tube diameter, turned out to be one-order more essential for the'

Author Response

To Reviewer #1: 

We would like to express special thanks to your valuable comments.

Ten corrections are suggested

.

[Q1]
 . Line024: A mistyping (letters) error; modify 'din ensity' to be:'than intrinsic material properties such as density, heat capacity, and electrical conductivity of the'

[A1] 'din ensity' is corrected to 'density'. 

[Q2]
 . Line108: Add the word "angular" in 'speeds' to be "angular speeds", to be:"roller. The 1st and the 2nd roller angular speeds, v1 and v2, were set up to 300 and 100 rpm, respectively, based"

[A2] 'speeds' is corrected to 'angular speeds'.

[Q3]
. Line131: Add the frequency (50 ? or 60 Hz) for the (experimental) electrical heating, in:"heating characteristics of the seamless heating tube, alternating current (?) was applied to the printer"

[A3] Thank you for your review for this. We applied DC instead of AC, which is mentioned in line 133.

[Q4]
. Line217: a) If you used Ni electrodes then modify the incompatible (Cu like) colour of the two-electrodes in Figure 2d.Line218: b) Correct (4->2) numbering reference: 'Figure 4d' to be "Figure 2d".

[A4] 'Ni electrodes' changes to 'copper electrodes' in the lines 211 and 218. 

[Q5]. Line250: Clarify, how did you measure the electrical conductivity, i.e. say, clearly, if it is a two-electrodes' configuration, DC or AC-conductivity (frequency= ? Hz):

[A5] Thank you for your review for this. 'alternating current' is changed to DC.

[Q6]. Line272: Add the Laser's frequency for the Raman spectroscopy, in: coating process, polarized Raman spectroscopy was measured and the results are presented in Figure 7

[A6] Wavelength for Raman spectroscopy is added in line 129-130 in the corrected version.

“with an excitation wavelength of 532 nm was used”

[Q7]
. Line320: Note/add the (temperature) error values in this temperature (180oC) range. Note errors of your temperature meter(s) taken from its specifications, in:"circumferential direction were 1.7, 2.3, and 3.0 oC, respectively. These correspond to +-0.65, +-0.8, and"

[A7] We add the standard deviation right next to the average temperature as follows

(Before correction) average temperature values were 135.7, 142.2, and 138.8 C

(After correction) average temperature values with standard deviation in the circumferential direction were 135.7 (0.6), 142.2 (0.7), and 138.8 (0.8) C.

[Q8]. Line326: If electrical heating is, actually, interrupted at time about 5.8s, then make it more clear, either adding a note in the text or/and the graph of Figure 8a.

[A8] We add a sentence in line 319-321 in the corrected version.

“The electrical heating was ceased at controller setting temperature of 200C, which resulted in maximum tube temperature of 210C at 5.8 sec, followed by temperature drop with air cooling.” 

[Q9]
 . Line343: Modify 'structural configuration' to be "morphological configuration"; so, it might be:'for the composite heating tube. From the analysis, the "morphological configuration" of the heating tube'

[A9] 'Structural configuration' changes to 'morphological configuration'.

[Q10]
 . Line344: Clarify more, how the conclusion wording 'one-order' comes from (which?) experiments, in:'such as composite thickness and tube diameter, turned out to be one-order more essential for the'

[A10] We add a sentence in line 348 in the corrected version.

The equation (4) describes the relationship between the heating performance geometrical parameters, and material property. From the theoretical analysis of temperature distribution on the cylinderical composite tube, as shown in Equation (4)”

Reviewer 2 Report

In the work titled “Flexible Seamless Tube-type Heater with Uniform Temperature Distribution and High Thermal Response Based on Carbon Nanotubes Aligned by Circumferential Shearing” and submitted by Y. Sohn et al., a flexible seamless tube‐type heater of MWCNTs/silicone rubber having uniform temperature distribution and fast thermal response has been developed by using a newly‐proposed circumferential shearing.

First of all, the title of the manuscript is too long and poor explanatory.

The abstract is too long; please, report only the part regarding the proposed novelty.

In the Experimental section, indicate who produces the used MWCNTs.

In the Experimental section, report references regarding the assertion reported at page 3 lines 93-94; how can the combination of the two materials reveal the reported properties? Please, discuss. Moreover, which is the reason in using MWCNTs with 100-200 nm in length and 10-20 nm in diameter? Discuss.

In the Experimental section, substitute “MWNTs” with “MWCNTs” at page 3 line 91.

Please, could you indicate the possible advantages in terms of cost in using this materials? Discuss.

Which is the process repeatability? Which is the material stability over the time? Discuss.

In my opinion, the manuscript is acceptable with minor revision.

In the manuscript, the English is quite acceptable.

Author Response

To Reviewer #2: 

We would like to express special thanks to your valuable comments.

Following is your comments and our answers are added.

----------------------------------------------------------------------------------------------------------

In the work titled “Flexible Seamless Tube-type Heater with Uniform Temperature Distribution and High Thermal Response Based on Carbon Nanotubes Aligned by Circumferential Shearing” and submitted by Y. Sohn et al., a flexible seamless tube-type heater of MWCNTs/silicone rubber having uniform temperature distribution and fast thermal response has been developed by using a newly-proposed circumferential shearing.

In my opinion, the manuscript is acceptable with minor revision.

In the manuscript, the English is quite acceptable.

Seven corrections are suggested.

----------------------------------------------------------------------------------------------------------

[Q1] . First of all, the title of the manuscript is too long and poor explanatory.

[A1] As-Is changes to To-Be. 

[As-Is] Flexible Seamless Tube-type Heater with Uniform Temperature Distribution and High Thermal Response Based on Carbon Nanotubes Aligned by Circumferential Shearing

[To-Be] Seamless Tube-type Heater with Uniform Thickness and Temperature Distribution Based on Carbon Nanotubes Aligned by Circumferential Shearing

[Q2]. The abstract is too long; please, report only the part regarding the proposed novelty.

[A2] Abstract is corrected, as follows.

The uniform temperature distribution, one of requirements for long-term durability, is essential for composite heaters. An analytical model for temperature distribution of a tube-type heater was derived, and it revealed that thickness uniformity is one-order more important than intrinsic material properties such as density, heat capacity, and electrical conductivity of the heating tube. In terms of characteristic dimension in axial and circumferential direction, the tube-type heater has a circumferential length shorter than an axial one. Based on the analytical model and on the consideration of the characteristic dimension, we introduced a circumferential shearing process to fabricate a flexible seamless tube-type heating layer of carbon nanotube/silicone rubber composite with outstanding uniform distribution of temperature and thickness, which may be attributed to shorter characteristic dimension in circumferential direction than in axial direction. The temperature uniformity was experimentally verified at various temperatures under heating. The difference in measured thickness and temperature in circumferential direction was within 1.3~3.0% (for tavg = 352.7 m) and 1.1% (for Tavg = 138.8 C), respectively, all over the heating tube. Therefore, the circumferential shearing process can be effective for cylindrical heaters, like a heating layer of a laser printer, which fuse toners onto papers during printing.

[Q3]. In the Experimental section, indicate who produces the used MWCNTs.

[A3] In line 90, “MWNTs, purchased from Hanhwa Chemicals (South Korea),” is added.

[Q4]. In the Experimental section, report references regarding the assertion reported at page 3 lines 93-94; how can the combination of the two materials reveal the reported properties? Please, discuss. Moreover, which is the reason in using MWCNTs with 100-200 nm in length and 10-20 nm in diameter? Discuss.

[A4] Reference number, [13], is added at the end of the sentence in lines 93.

[Q5]. In the Experimental section, substitute “MWNTs” with “MWCNTs” at page 3 line 91.

[A5] The abbreviation of multi-walled carbon nanotubes is unified to “MWNTs” in the whole manuscript.

[Q6]. Please, could you indicate the possible advantages in terms of cost in using this materials? Discuss.

[A6] Cost can be discussed in materials and process. Cost of circumferential shearing-based CNT fusers will be comparable to that of extrusion-based CNT fusers. Cost of both material and process should be considered, but our experiment is a lab-scale based process, and so we are careful to make a comment on cost at this current stage. 

[Q7]. Which is the process repeatability? Which is the material stability over the time? Discuss.

[A7] In order to mention about process repeatability, we need to get data about the process in a “pilot” basis. Our study was done in a lab basis. Therefore, we need to move forward to pilot-based study, but we didn't yet. Nevertheless, our data in the manuscript have a good process repeatabiliby for ten specimens.  

We add a sentence, “leading to enhancement in process repeatability” in line 77. 

For the material stabiliy, two additional references [14, 15] having TGA data which showed material stability of nanocarbon (graphene [14] and MWNT [15]) with PDMS used in this study.

We add a sentence in line 93. “and also for material stability at least up to 230~250C [14-16].”